# Advanced Millimeter-Wave Radar System for Real-Time Multiple-Human Tracking and Fall Detection

**DOI:** 10.3390/s24113660

**Published:** 2024-06-05

**Authors:** Zichao Shen, Jose Nunez-Yanez, Naim Dahnoun

**Affiliations:** 1School of Electrical, Electronic and Mechanical Engineering, University of Bristol, Bristol BS8 1UB, UK; 2Department of Electrical Engineering, Linköping University, 581 83 Linköping, Sweden; jose.nunez-yanez@liu.se

**Keywords:** millimeter-wave radar, multiple-target tracking, human fall detection, human activity recognition (HAR), real-time processing, internet of things (IoT) application

## Abstract

This study explored an indoor system for tracking multiple humans and detecting falls, employing three Millimeter-Wave radars from Texas Instruments. Compared to wearables and camera methods, Millimeter-Wave radar is not plagued by mobility inconveniences, lighting conditions, or privacy issues. We conducted an initial evaluation of radar characteristics, covering aspects such as interference between radars and coverage area. Then, we established a real-time framework to integrate signals received from these radars, allowing us to track the position and body status of human targets non-intrusively. Additionally, we introduced innovative strategies, including dynamic Density-Based Spatial Clustering of Applications with Noise (DBSCAN) clustering based on signal SNR levels, a probability matrix for enhanced target tracking, target status prediction for fall detection, and a feedback loop for noise reduction. We conducted an extensive evaluation using over 300 min of data, which equated to approximately 360,000 frames. Our prototype system exhibited a remarkable performance, achieving a precision of 98.9% for tracking a single target and 96.5% and 94.0% for tracking two and three targets in human-tracking scenarios, respectively. Moreover, in the field of human fall detection, the system demonstrates a high accuracy rate of 96.3%, underscoring its effectiveness in distinguishing falls from other statuses.

## 1. Introduction

Human activity recognition (HAR) systems have garnered significant attention in the industry, particularly in the field of camera-based systems leveraging machine learning techniques [1,2,3,4]. However, these camera-based systems come with drawbacks, including privacy invasion, dependency on specific lighting conditions, and reduced performances in the presence of smoke or fog. As a solution to these challenges, many researchers are turning to Millimeter-Wave (mmWave) radar technology employing the Frequency Modulated Continuous Wave (FMCW) technique.

The mmWave radar operates at a high frequency range (from 76 to 81 GHz), providing several advantages such as high resolution and improved anti-interference capabilities. Consequently, FMCW mmWave radar technology has demonstrated significant potential in various indoor HAR applications, including posture detection [5,6,7] and human identification [8]. Furthermore, human tracking and fall detection represent popular applications for mmWave radar, addressing critical safety concerns in various settings such as the healthcare system for the elderly. Many researchers have proposed numerous mmWave radar systems for human tracking as well [8,9,10].

In this paper, we delve into the application of mmWave radar for human tracking and fall detection, covering the operational principles, ongoing research, and development efforts. Additionally, we present a real-time system and elaborate on how it successfully accomplishes its objectives. Leveraging three FMCW radar IWR1843 development boards from Texas Instruments (TI), we enhanced the precision of human tracking and fall detection. Consequently, our system delivers accurate real-time results for multiple human targets in indoor environments. The primary contributions of our work include the following:We deployed three radars to expand the coverage area and designed a real-time system that collaborates with all sensors to capture point clouds at 20 frames per second (FPS) from a scene.We introduced innovative strategies, including dynamic Density-Based Spatial Clustering of Applications with Noise (DBSCAN) for enhanced target detection when the human target is static, a probability matrix for multiple-target tracking, and target status prediction for fall detection.We assessed our system through over 300 min of experimentation covering single- and multi-person scenarios with walking, sitting, and falling actions, demonstrating its performance in both human target tracking and fall detection.We made our work open-source at https://github.com/DarkSZChao (accessed on 10 March 2024) to further promote work in this field.

The remaining sections of this paper are organized as follows. Section 2 provides a brief overview of the principles and related works concerning mmWave radar. In Section 3, we discuss the evaluation of the mmWave radar system, covering aspects such as angle of view compensation and the relationship between radar placement and coverage. Section 4 illustrates the mmWave radar setup and data collection. Subsequently, Section 5 delves into the details of our software framework’s workflow and its utilization for human tracking and fall detection. We present an evaluation of our real-time system performance in scenarios involving multiple people in Section 6. Finally, Section 7 outlines our conclusions and discusses avenues for future work.

## 2. Background and Related Work

In this section, we present an overview of current state-of-the-art mmWave radars using the FMCW technique and novel applications of this hardware for human tracking and fall detection.

### 2.1. Tracking and Fall Detection Approaches

Prevalent tracking and fall detection methods can be categorized into two approaches: wearable and non-wearable solutions. Wearable devices, incorporating sensors like inertial measurement units, accelerometers, and gyroscopes on the human body, as proposed in [11,12,13], enable the fusion of sensor data for tracking and fall detection and the safeguarding of individuals. However, wearable devices pose inconveniences for people, especially the elderly with poor memory and limited mobility. To address this issue, non-wearable camera-based fall detection systems, as suggested in [3,4], have been adopted, leveraging deep learning and background subtraction techniques for indoor environments. While these systems yield accurate results across various distances, they face challenges related to privacy concerns due to camera intrusiveness and limitations arising from lighting conditions.

To address challenges related to the intrusiveness and lighting limitations faced by camera-based fall detection systems, many researchers have shifted their focus to detecting human bodies using mmWave radars [9,14,15]. Typically, mmWave radar is deployed in scenarios demanding higher accuracy due to the use of short-wavelength electromagnetic waves. Beyond its successful application in autonomous driving, mmWave radar has been employed in the field of human tracking and fall detection. A recent real-time human detection and tracking system using Texas Instruments (TI) mmWave radars was established in [9]. The authors introduced a software framework capable of communicating with multiple radars, consistently achieving over 90.4% sensitivity for human detection in an indoor environment. Subsequently, the research in [5] delved into human posture, presenting an analysis report on the capabilities of mmWave radar in human recognition. Building on this analysis, ref. [7] merged mmWave radars with the Convolutional Neural Network (CNN) technique to accurately estimate human postures with an average precision of 71.3%. Moreover, refs. [6,16] implemented a CNN for point cloud analysis to estimate human skeletons and the postures of patients. In contrast, refs. [17,18] focused more on outdoor environments, proposing fusion systems incorporating both mmWave radar and camera methods for object detection and tracking. Our work, inspired by the human detection system in [9,14], deployed three IWR1843 mmWave radars on the x-y-z surfaces concurrently to capture more robust human body reflection signals. Additionally, we established a concurrent real-time system to track humans and classify the target status.

### 2.2. MmWave Radar Preliminaries

This section provides a concise overview of mmWave radar theory, with more comprehensive details available in [19]. For our experiments on human tracking and fall detection, we employed IWR1843 FMCW mmWave radars developed by Texas Instruments (TI), operating at a frequency range of 76–81 GHz with a maximum available bandwidth of 4 GHz. This radar development board features three transmitters (TX) and four receivers (RX), resulting in twelve virtual antennas operating simultaneously [20,21] (see Figure 1).

With the FMCW technique, the mmWave radar can transmit chirp signals ($S_{tx}$) with a continuously varying frequency within the bandwidth. The reflected signal ($S_{rx}$) is collected and mixed with $S_{tx}$ to generate an intermediate frequency (*IF*) signal, as illustrated in Figure 2. The frequency and phase of the *IF* signal correspond to the difference between $S_{tx}$ and $S_{rx}$. In utilizing Cortex-R4F and C674x DSP chips, a data processing chain is then applied to the *IF* signal to create a Range-Doppler Map (RDM) using Fast Fourier transforms (FFTs). Subsequently, the Constant False Alarm Rate (CFAR) algorithm was employed to identify peaks by estimating the energy strength on the RDM [22].

#### 2.2.1. Distance Measurement

The target distance can be calculated using Equation (Equation 1), where *S* represents the slope rate of the transmitted chirp, and τ is the time of flight. The time of flight (τ) is the round-trip distance (2d) divided by the speed of light (*c*). Consequently, we can estimate the target distance (*d*) as follows:(1)$$f_{IF}=S\tau =\frac{S\cdot 2d}{c}\Rightarrow d=\frac{f_{IF}c}{2S}$$

#### 2.2.2. Velocity Measurement

To ascertain the target velocity, the radar emits two chirps separated by time $T_{c}$. Each reflected chirp undergoes FFT processing to identify the range of the target (Range-FFT). The Range-FFT corresponding to each chirp exhibits peaks in the same locations but with different phases. The observed phase difference corresponds to motion in the target of $vT_{c}$. The velocity *v* can be computed using Equation (Equation 2), where Δϕ represents the phase difference:(2)$$\Delta \phi =2\pi \frac{\Delta d}{\lambda }=2\pi \frac{2vT_{c}}{\lambda }\Rightarrow v=\frac{\lambda \Delta \phi }{4\pi T_{c}}$$

For precise velocity estimation, the radar transmits multiple consecutive chirps to create a chirp frame. Subsequently, it conducts a Doppler-FFT over the phases received from these chirps to determine the velocity.

#### 2.2.3. Angle Measurement

The configuration of multiple transmitters (TX) and receivers (RX), as depicted in Figure 1, introduces a variance in distance between the target and each antenna, causing a phase shift in the peak of the Range-FFT or Doppler-FFT. The FMCW radar system leverages this phase difference between two chirps received by RX modules to calculate the angle of arrival (AoA) for the target, as illustrated in Figure 3.

Assuming a known RX antenna with a spacing of l=λ/2 (Figure 1), we can determine the AoA (θ) from the measured Δϕ using Equation (Equation 3):(3)$$\Delta \phi =2\pi \frac{\Delta d}{\lambda }=2\pi \frac{l\sin(\theta )}{\lambda }\Rightarrow \theta =\sin^{-1}\left(\frac{\lambda \Delta \phi }{2\pi l}\right)$$

Finally, the Arm Cortex-R4F-based radio control system on board will provide the point cloud of the targets and send it to the PC for further processing.

## 3. Radar System Evaluation

### 3.1. Multiple Radar Arrangement

The theoretical Angle of View (AoV) for mmWave radars is ±90° in both horizontal and vertical directions. However, according to the radar evaluation manual [21], the effective AoV for a 6 dB beamwidth is reduced to around ±50° horizontally and ±20° vertically due to antenna characteristics and signal attenuation. This reduced vertical AoV means that a radar placed at a height of 1 m can only capture signals reflected from the human chest to the knee, limiting its ability for complete human body detection.

To address this limitation, we deployed three identical TI IWR1843 mmWave radars on both the wall and ceiling. This setup allowed us to capture strong signals not only from the human main body but also from the human head. For detailed information, please refer to Section 4.

When employing multiple radars, it is crucial to ensure that they do not interfere with each other. If we consider a maximum measurement distance of 4 m for example, the round-trip time-of-flight would be 0.027 μs. Given a slope rate of 70 MHz/μs, this time interval corresponds to a frequency change of approximately 1.87 MHz, as illustrated in Figure 4.

Referring to the interference probability equation (Equation (Equation 4)) for *N* radars presented in [9], we can compute the interference probability for our scenario involving three radars. In this equation, $B_{total}$ represents the 4 GHz chirp bandwidth of the TI radar, and $B_{inter}$ denotes the interference bandwidth. This implies that radars will only interfere if the frequency difference between any two radars falls within the 5.6 MHz range for our experiment [9]. Ultimately, the interference probability for three radars is 0.4%, assuming that the radars are activated at random times.
(4)$$P\left(N\right)=1-\prod_{i=1}^{N}\frac{B_{total}-B_{inter}\cdot \left(i-1\right)}{B_{total}}$$

Additionally, ref. [9] demonstrated that the average variances of radar detection with and without interference from a second radar are quite similar, even when the two radars are within a 3 m range. Consequently, we can conclude that the interference between radars for our experiment is minimal and can be disregarded.

### 3.2. Radar Placement and Coverage Evaluation

In our previous study [23], we positioned the radars at a height of 1 m on the wall, placed perpendicular to the ground, thereby ensuring comprehensive coverage from the knee to neck level. Despite these efforts, this arrangement yielded a suboptimal resolution and restricted recognition capabilities when human subjects approached the wall (radar) closely. Specifically, as depicted in Figure 5, this setup led to a diminished coverage area of approximately 50% when the human target approached within 1 m of the radar. This limitation arose due to the inadequacy of the mmWave signal emitted from the radar (green) positioned at a height of 1 m, failing to sufficiently spread across the human body’s main area (e.g., head) before encountering reflections.

To address this limitation, we chose to move the radar higher and tilt it at a specific angle (55° was optimal in our experiment) to point it toward the center of the ground, as the radar (red) shown in Figure 5. This alteration aimed at broadening the mmWave signal’s coverage, allowing it to encompass a larger area of the human body before reflection. It is worthy to note that the angle and height configurations can be adjusted for rooms of different sizes and heights to achieve optimal performance.

Figure 6 presents 200 frames accumulated over a span of 10 s, depicting a scenario where a person stands stationary approximately 1 m away from the wall. In the left column, the point cloud received and the corresponding histogram for the radar positioned at a height of 1 m is displayed. Most data points cluster tightly in the middle of the human target due to the radar’s limited field of vision, leading to the loss of information regarding the head and legs. Consequently, the system might interpret this as a seated human due to the absence of data on the head and chest [23].

However, the point clouds and histograms shown in the middle and right columns, captured by the radars placed at a height of 2.6 m (tilt of 55° and 50°), reveal a significant concentration of points representing the human head, chest, abdomen, and legs. These comprehensive data ensure the accurate classification of the human’s status, even when the person is close to the wall, eliminating any potential blind spots. In our experiment, the coverage provided by the 50° tilt angle is relatively poor compared to the 55° tilt, as it less effectively covers the human body parts below the chest. Meanwhile, a larger tilt angle causes the radar to face more toward the ground, leading to decreased detection ability when the person is relatively far from the wall. Therefore, we chose the 55° angle for the setup.

## 4. Experimental Setup

### 4.1. Radar Characterisation

To generate 3D point cloud data, we configured three IWR1843 radars to utilize all three TXs and four RXs following the instructions of the TI mmWave visualizer [24]. The start frequency and end frequency were set to 77 GHz and 81 GHz, respectively, resulting in a bandwidth (*B*) of 4 GHz. The Chirp Cycle Time ($T_{c}$), defined as the duration of each sweep cycle, was set to 162.14 μs, while the Frequency Slope (*S*), representing the sweep rate of the FMCW signal, was set to 70 MHz/μs. With this setup, the sensor achieved a range resolution of 4.4 cm and a maximum unambiguous range of 5.01 m. Regarding velocity, it could measure a maximum radial velocity of 2 m/s with a resolution of 0.26 m/s. This level of measurement accuracy was sufficient for capturing human movements within the room, supporting our identification project. To balance the workload between the three radars and the PC, we configured each radar to operate at 20 frames per second, with 32 chirps for each frame.

The experimental space occupied a space with dimensions of 4 m in width, 4.2 m in length, and 2.6 m in height. Within this space, Radar 1 was vertically positioned on a desk adjacent to a wall, Radar 2 was installed on a wall with a 55° tilt, and Radar 3 was mounted on the ceiling, directed toward the ground, as depicted in Figure 7. All three radars were connected to a PC through USB cables and extensions. Additionally, a camera was strategically placed at the upper corner of the room to serve as a visual reference. Human targets could enter the monitored area through a door.

### 4.2. Data Collection

We collected point clouds from 15 randomly selected individuals, resulting in a total duration of approximately 300 min (∼360,000 frames). Initially, each participant was instructed to walk around the radar room for 10 min, constituting 51.5% of the total duration (156 min) when only one person was present in the scene. Subsequently, participants formed randomly assigned pairs and entered the scene for 10 min intervals, for a total of 10 rounds, accounting for 33.1% of the total duration (100.4 min) when two people were present. Similarly, for scenarios involving three individuals, groups of three entered the scene for around 5 min per round, totaling 10 rounds, which constituted 15.4% of the total duration (46.7 min). In all scenarios, participants were instructed to sit on a chair or fall to the ground randomly throughout the duration of the experiment. The data collection composition is shown in Table 1. Since we evaluated the data target by target, the duration was multiplied by 2 and 3 for two-target and three-target scenarios, respectively, to obtain the total duration for each target.

The participants, aged 20–35, varied in height (160–185 cm) and weight (50–95 kg), encompassing both male and female representations. The video footage was only used to verify the system’s performance. This data collection received approval from the Faculty of Engineering Research Ethics Committee at the University of Bristol (ethics approval reference code 17605).

To maintain consistency, all point clouds were represented in the coordinate format (x,y,z,v,snr), where (x,y,z) denotes the coordinates of the point, and (v,snr) signifies the velocity and Signal-to-Noise Ratio (SNR) level, respectively. A global coordinate system was established to facilitate the alignment of points gathered from different radars through spatial rotation and positional adjustment. Since radars output point clouds of arbitrary sizes, the resulting dataset was structured as N×P×C, where *N* represents the number of samples (frames), *P* indicates the number of points in each frame (of arbitrary size), and *C* denotes the number of feature channels (C=5). The elements of this tensor can be represented as $X_{n,p,c}$, where n,p, and *c* correspond to the respective indices.
(5)$$X\in R^{N\times P\times C}$$

## 5. System Design

Our system was developed using Python 3.8 and relies on essential libraries like NumPy, Sklearn, and Matplotlib. It operates on a Desktop PC featuring an Intel(R) Core(TM) i7-10850H CPU @ 2.70 GHz and 16 GB of RAM. This system works under 20 FPS. Figure 8 illustrates the system’s architecture, which comprises the following main modules:*Data_Reader*: Data Readers parse the data packages sent by the radars.*EProcessor*: The Early Processor provides data rotation and position compensation based on the radar placement.*PProcessor*: The Post Processor provides data filtering, clustering, and target tracking.*Visualizer*: The Visualizer provides 3D demonstrations for the human tracking and status.*Queue_Monitor*: The Queue Monitor monitors the frame traffic and provides synchronization.*Camera*: The Camera provides video footage during the experiment as ground truth.

### 5.1. Radar Raw Data Preprocessing

Due to the limited number of data points collected in each frame, we aggregate 10 frames to create a frame group for processing at that moment. A sliding window approach, spanning 10 frames (equivalent to 0.5 s), is employed to shift the frames forward, ensuring continuous processing.

To account for the utilization of multiple radars positioned at varying locations and angles, compensation methods for data point rotation and repositioning are applied.

#### 5.1.1. Rotation Compensation

Three 3D rotation equations, Equations (Equation 6)–(Equation 8), for rotation compensation are presented below [25,26,27]:(6)$$RM_{x}=\left[\begin{array}{cccc} 1 & 0 & 0 & 0\\ 0 & \cos\alpha & -\sin\alpha & rp_{y}\left(1-\cos\alpha \right)+rp_{z}\sin\alpha \\ 0 & \sin\alpha & \cos\alpha & rp_{z}\left(1-\cos\alpha \right)-rp_{y}\sin\alpha \\ 0 & 0 & 0 & 1 \end{array}\right]$$
(7)$$RM_{y}=\left[\begin{array}{cccc} \cos\beta & 0 & \sin\beta & rp_{x}\left(1-\cos\beta \right)-rp_{z}\sin\beta \\ 0 & 1 & 0 & 0\\ -\sin\beta & 0 & \cos\beta & rp_{z}\left(1-\cos\beta \right)+rp_{x}\sin\beta \\ 0 & 0 & 0 & 1 \end{array}\right]$$
(8)$$RM_{z}=\left[\begin{array}{cccc} \cos\gamma & -\sin\gamma & 0 & rp_{x}\left(1-\cos\gamma \right)+rp_{y}\sin\gamma \\ \sin\gamma & \cos\gamma & 0 & rp_{y}\left(1-\cos\gamma \right)-rp_{x}\sin\gamma \\ 0 & 0 & 1 & 0\\ 0 & 0 & 0 & 1 \end{array}\right]$$
(9)$$P=\left[\begin{array}{ccc} P1_{x} & \; & Pn_{x}\\ P1_{y} & ... & Pn_{y}\\ P1_{z} & \; & Pn_{z}\\ 1 & \; & 1 \end{array}\right]\;P^{\prime }=\left[\begin{array}{ccc} P1_{x}^{\prime } & \; & Pn_{x}^{\prime }\\ P1_{y}^{\prime } & ... & Pn_{y}^{\prime }\\ P1_{z}^{\prime } & \; & Pn_{z}^{\prime }\\ 1 & \; & 1 \end{array}\right]$$

In the equations provided, $RM_{x}$, $RM_{y}$, and $RM_{z}$ represent rotation transformation matrices along the x, y, and z axes, respectively. *P* and $P^{\prime }$ (Equation (Equation 9)) denote the point matrices before and after transformation. The rotation angles for the x, y, and z axes, denoted by α, β, and γ, respectively, are determined by the radar facing angles. The reference point coordinates in 3D, represented as $rp_{x}$, $rp_{y}$, and $rp_{z}$, are set to (0,0,0), indicating the origin in our experiment. In using Equation (Equation 10), the data points obtained from the radars can be transformed and mapped into a unified global coordinate system.
(10)$$P^{\prime }=RM_{x}RM_{y}RM_{z}P$$

#### 5.1.2. Position Compensation

For the position compensation, Equation (Equation 11) is applied to correct the radar position offset in the global coordinate system.
(11)$$P^{\prime }=\left[\begin{array}{cccc} 1 & 0 & 0 & \Delta x\\ 0 & 1 & 0 & \Delta y\\ 0 & 0 & 1 & \Delta z\\ 0 & 0 & 0 & 1 \end{array}\right]P$$

After applying point repositioning and rotation algorithms based on the radars’ positions and facing directions, the results are then placed into *queues*, where they await further processing and analysis.

### 5.2. Multiple Radar Data Line Synchronization

All raw data sent by a TI mmWave radar comprising coordinates and speed information are correctly packaged into frames and follow the prescribed TI output packet structure. Occasionally, the hardware may send incomplete packets that cannot be parsed into usable data for the following process and interrupt the sequence.

To address this issue, we implemented a timestamp for all frame packages and introduced a *Queue_Monitor* module designed to oversee the packet accumulation within the *queues*. We established the following criterion: Frame packages received from three radars within a 0.05 s window (20 FPS) are considered synchronized, and we merge them before sending them to the *PProcessor*. Subsequent packages will be aggregated in the next 0.05 s time period, as depicted in Figure 9.

Additionally, the implementation of the FIFO queue strategy enhances the system’s ability to process data in real time. In situations where there is data congestion within the *queues*, the *PProcessor* ensures that it retrieves the earliest data according to the established timeline. This approach guarantees sequential processing, even in worst-case scenarios, thereby ensuring the system’s responsiveness.

### 5.3. Background Noise Reduction

After data retrieval from each radar by the *PProcessor*, shown in Figure 8, the data points with low SNR are identified and stored by the *Global BES_Filter* module for background recognition. Additionally, noise points identified by the *Dynamic DBSCAN* module are also collected for background recognition in the subsequent process. In utilizing the information provided by both the *Global BES_Filter* and *Dynamic DBSCAN*, the *BGN_Filter* module effectively discerns background noise and isolates areas where data points marked as noise persist for an extended duration. This is especially applicable to cases where static targets such as chairs and tables are present in the field. This feedback loop approach for noise reduction results in a clean frame comprising data points with reduced noise being generated and utilized for DBSCAN clustering.

### 5.4. Human Target Detection

In the process of identifying clusters representing human beings amidst the noise, we employ the *Dynamic DBSCAN* module for the DBSCAN algorithm. DBSCAN stands out due to its ability to function without the need for specifying the number of clusters in advance, making it particularly suitable for situations where the exact number of human targets is unknown [28]. This algorithm groups closely packed data points into dense regions, differentiating them from sparser areas. Such an approach effectively addresses scenarios where dynamic targets are encountered in our experiment.

However, we observed that when a human target is stable, fewer data points with a high SNR level are collected due to the mmWave radar’s limited sensitivity to stationary targets, a finding supported in [9]. Because stationary targets cannot be effectively distinguished from background noise in the Range-Doppler Map (RDM) [29], they are consequently treated as noise and removed in our analysis.

Although we cannot directly address the challenge of distinguishing stationary targets, an inherent characteristic of mmWave radar, we enhanced accuracy with a novel dynamic DBSCAN approach, which is a multiple-level approach, to cluster the data points rather than inputting all data points directly into the DBSCAN algorithm. The primary advantage of this approach is the prevention of missing valuable data points with high SNR levels, which are highly likely to represent human targets instead of noise.

We first categorize the points based on their SNR values. Subsequently, we apply more lenient DBSCAN parameters to points with a higher SNR and stricter parameters to the low-SNR points, as shown in Figure 10. For instance, if we were to treat all points equally and apply default parameter settings of ε=0.5 and minPts=10, where ε denotes point distance and minPts is the minimum number of points used by DBSCAN to define a cluster, the cluster would not form due to an insufficient number of points, as shown in the left image of Figure 11. In this case, the high-energy points representing the human chest, which should be identified as a cluster, are missed. To address this, as an example, we employed more lenient parameters: ε=1 and minPts=2 for points with higher SNR levels above 300, and ε=0.7 and minPts=3 for points with SNR levels above 200. These settings allow us to successfully form a cluster representing a stationary human target (see the right side of Figure 11). It is worth noting that these parameter settings are selected based on the point cloud density observed in our experiments. These settings can be adjusted if a different number of radars are used or for rooms of varying sizes, as the radars will produce point clouds with different point densities. Employing dynamic DBSCAN necessitates five times the computation for each frame clustering in the worst-case scenario, wherein data points in one frame span all predefined SNR regions in Figure 10. However, such a scenario is infrequent. Typically, most data points fall within a single SNR region when no human presence is detected, and only two to three regions are occupied when humans are present. Given this observation, we made an optimization decision. If a specific SNR region contains no data points, we choose to skip the corresponding DBSCAN process for that region to conserve computational resources and enhance the processing speed by avoiding unnecessary computations.

### 5.5. Human Target Tracking

To achieve the real-time tracking of the position, status, and moving trajectory of detected human targets, we provided the *Obj_Status Bin* modules, designed specifically to store this information. With multiple clusters generated by the *Dynamic DBSCAN* module, the issue lies in accurately assessing these clusters and assigning them to the corresponding *Obj_Status Bin* modules for continuous tracking.

To address this challenge, we introduced the *TrackingSys* module, outlined in Figure 8. This module serves the crucial role of determining which clusters from the *Dynamic DBSCAN* module should be allocated to which *Obj_Status Bin* modules based on a probability matrix. Our strategy of a probability matrix is generated by evaluating the correlation between each potential cluster and the previous information stored in the *Obj_Status Bin* modules, i.e., the previous cluster position and shape. The elements of the probability matrix are determined using the following equation comprising four components:(12)$$P_{clu}=\alpha C_{pos}+\beta C_{shape}+\gamma E_{pos}+\delta E_{shape}$$

In this equation, $C_{pos}$ represents the correlation factor between the position of the potential cluster and the previously stored positions in the *Obj_Status Bin* modules, while $C_{shape}$ indicates the correlation for the cluster shape. Specifically, we used the Z-Score to evaluate these relationships and identify outliers that should be disregarded. $E_{pos}$ and $E_{shape}$ denote the position and shape difference between the potential cluster and the predefined expected values. For example, we do not anticipate a cluster located at the ceiling to be identified as a human being. Additionally, we used proportion coefficients [0.3,0.3,0.2,0.2] for our experiment to balance these four components and calculate a more accurate $P_{clu}$. These coefficients can be adjusted to adapt the system to various deployment environments, ensuring flexibility and accuracy in the tracking process.

After applying Equation (Equation 12) to each current cluster for every *Obj_Status Bin* module, the *TrackingSys* module generates the probability matrix. Figure 12 demonstrates an example of a two-people scenario. The probabilities between each potential cluster and object bins are produced. The module utilizes the global maximum probability, highlighted in red in Figure 12, to update the cluster to the corresponding object bin. Subsequently, this cluster and its neighbouring clusters are regarded as the same person and removed from consideration. The process is then repeated: the next maximum value from the remaining possibilities is selected and allocated, continuing until there are no non-zero values left in the probability matrix.

Ultimately, the *Obj_Status Bin* modules provide human tracking and status information to the *Visualizer*, facilitating the display of human target positions and shapes based on historical cluster data. This method ensures the accurate and real-time tracking of human targets within the environment.

### 5.6. Fall Detection

This study, which lies solely on fall detection for individuals who require medical surveillance at home or hospital, excluded other postures considered in [15,30], simplifying the task. This objective eliminates the necessity for neural networks to solve the problem. By avoiding computationally expensive algorithms like neural networks, which demand additional Graphic Processing Units (GPUs) and entail much higher computation costs and power consumption, our system becomes easily deployable on edge devices and low-power consumption platforms with embedded processors in the future for IoT applications.

To determine the current status (walking, sitting, or lying on the ground) of a human target, we predefined estimated portraits of position and shape for these three statuses. For example, if the target is walking, the center height of the cluster representing this target is estimated at around 1 m. Additionally, the cluster shape is modeled as a rectangle cuboid with its long side aligned with the z axis. When the target is sitting, the center height is estimated to be around 0.6 m. If the target is lying on the ground, the center height is approximately 0.2 m, close to the ground level. In this case, the cluster shape is a rectangle cuboid with its short side on the z axis. To calculate the status probability, we update Equation (Equation 12) to Equation (Equation 13), as shown below:(13)$$P_{sta}=\lambda E_{pos}+\sigma E_{shape}$$

$E_{pos}$ and $E_{shape}$ represent the position and shape difference between the cluster and the predefined portraits. To balance these factors, we used proportion coefficients [0.7,0.3] in our experiment. Following this calculation, the status probability for each cluster assigned to the *Obj_Status Bin* in Section 5.5 is computed. The cluster is then labeled with the status with the highest probability.

To enhance the stability of the determined target status, we employed a blur process. This process prevents the status from being updated to a new state if there are not enough clusters indicating the new status. As illustrated in Figure 13, we use a sliding window of a certain length (20 frames in our experiment) along the stored clusters in each *Obj_Status Bin*. The target status is determined based on which status has the largest number of clusters within the sliding window.

Additionally, we developed a notification module named *Gmail_Notifier* based on the Gmail API [31]. This module is intended to send alerts in case of a detected human fall, as depicted in Figure 8.

## 6. System Evaluation

To assess our real-time system’s performance, we analyzed frames with a total length of 300 min (details can be found in Section 4.2). Meanwhile, we took a video recording as the ground truth by placing a camera at the top corner of the experimental field. We utilized the Yolo-v3 model [32] to obtain the ground truth, establishing a baseline for human detection evaluation.

### 6.1. Multiple-Human Tracking Evaluation

We specified that a successful detection requires the centroids of the human target labeled by both our system and the camera detection to be within 0.25 m of each other, with a minimum overlapping frame area of 70%, as outlined in [9]. The following metrics were employed for the evaluation of our human tracking system:*Positives (P)*: Humans are present in the experimental field.*True Positives (TP)*: Humans are present and all identified by the radar, with their positions verified by the camera.*False Positives (FP)*: Humans are absent and identified by the radar caused by noise or other objects, or their positions are not verified by the camera.*Sensitivity (TP/P*): The ability to identify humans with valid positions when they are present in the detection area.*Precision (TP/(TP+FP))*: The ability to identify humans with valid positions from false detection caused by noise.

Leveraging three radars in the field and our techniques, we achieved, approximately, a 98% sensitivity and 98.9% precision for scenarios with a single target in the field, as shown in Table 2. This implies that our system can effectively track a human target when it appears and exhibits a strong ability to distinguish it from noise. For scenarios involving two and three people in the field, the sensitivity remains high, indicating that our system can easily detect the presence of targets. However, the positions may not be accurate when multiple targets are present. The precision experiences slight drops of 2.4% and 4.9% for the two-person and three-person scenarios, respectively, which proves this phenomenon. This decline is attributed to the increased presence of moving people in the field, leading to more noise and false detections at incorrect positions. The average F1 score of our tracking system is 97.2%.

Figure 14 displays three examples featuring two and three mobile human targets in the given environment. In the images on the left column, the top views of the point cloud are presented. The three radars are denoted as dark red dots, while the dots with varying shades represent the point cloud with different SNR levels. The DBSCAN clusters are depicted with red rectangles. The middle-column images show the trajectories of the multiple human targets’ movements in the field, labeled with green, purple, and sky blue, respectively. The corresponding ground truth, captured by the camera, is displayed in the images in the right column.

A limitation of our radar system is the challenge of distinguishing multiple people at short distances, particularly when individuals are walking close to each other. This limitation is less pronounced under typical normal conditions when individuals are usually separated. While [9] achieved success at distances of 1 m, we observed the system’s distinguishability to be 0.5 m when targets are in motion and 0.3 m when targets are stationary.

### 6.2. Human Fall Detection Evaluation

We evaluated the human status using the data classified as True Positives (TP) in the last Section 6.1. Figure 15 illustrates the confusion matrix for the three statuses we classified: walking, sitting, and fall detected (lying or sitting on the ground). We achieved a high sensitivity level of 99.0% for fall detection and 97.7% for walking. Although we observed a relatively lower sensitivity of 92.6% when the target is sitting, it is noteworthy that most of the false cases are classified as walking rather than fall detection. This indicates that, while our system has approximately a 2% to 7% chance of misjudgment between walking and sitting statuses, it effectively distinguishes fall detection from the other two statuses. This is proved by the average F1 score of 96.7% across all three categories, with a particularly high F1 score of 99.5% for fall detection. Our system boasts an average precision of 97.1% and sensitivity of 96.4%, with an overall accuracy rate of 96.3%.

Figure 16 displays a 3D point cloud for when a human target falls in the left column. The middle column shows the moving trajectories with the target status marked in green (walking), yellow (sitting), and red (fall detected) dots. The camera image serves as the ground truth. When human targets fall and lie on the ground, the point clouds are concentrated at a lower height, allowing them to be identified as a fall on the ground using the method introduced in Section 5.6.

The presence of yellow dots in the trajectories of the second and third examples does not signify that the targets were sitting. Rather, these dots appear because the targets simulated falling to the ground during our experiments. The fall speed was intentionally slow, not corresponding to a real fall, which resulted in not skipping the sitting status. Sitting is considered a transitional state between walking and falling on the ground (lying). Therefore, the presence of yellow dots in the trajectories captured by our system is a normal outcome. In the third example, our system detects a fall when there are multiple people in the room. Because the second human target remained walking throughout the observation period, to distinguish this trajectory from the first target, we mark it as purple in the visualization.

A limitation of our fall detection system is the challenge of distinguishing between an actual fall and someone intentionally lying down on a mattress. If a person lies on a bed with some height, it remains relatively easy to differentiate from a fall based on the z axis coordinates. However, if the person lies down on a mattress, we must rely on the speed of the event to make a distinction. For instance, a longer duration of the event can be interpreted as going to sleep on a mattress, whereas an instant duration may indicate a fall. Further investigation into this aspect is left for future work.

### 6.3. Human Fall Posture Estimation

Fall posture estimation is a crucial step following successful fall detection, especially for elderly individuals. Knowing the posture during a fall can help healthcare providers assess the severity of the fall and provide appropriate medical intervention.

We assumed that the person remains relatively stationary after falling. To determine the posture after a fall, we accumulated point clouds over a period of 30 s and analyze the resulting data points. As illustrated in Figure 17, point clouds from both the top view and side view, as well as camera images, were compared for three different fall scenarios: lying facing up, lying facing sideways, and sitting on the ground.

In the scenario where the person is lying facing up, the top view exhibits the largest reflection area and the lowest gathering of point clouds, occurring approximately 0.25 m above the ground. In the second scenario, where the person is lying facing sideways, the top view shows a narrower area, and the gathering of points occurs at a higher position around 0.5 m in the side view. Finally, in the scenario where the person is sitting on the ground, the top view displays the smallest area, and points gather around 1 m above the ground, representing the human head.

Hence, it is viable to assess and categorize typical human fall postures by analyzing the point clouds collected using mmWave radars. Further exploration may involve the introduction of additional posture classes and real-time estimation as part of future work.

### 6.4. System Comparison

We present a comparative analysis of human tracking and fall detection approaches in Table 3. Although wearables [12], and the camera-based [4] methods demonstrated high accuracy in fall detection, they encountered challenges related to inconveniences in deployment and severe privacy issues. In contrast, mmWave radar solutions do not experience these issues. Approaches from [15,30,33] achieved over 92% in accuracy in fall detection, but they lack support for human tracking and for scenarios involving multiple individuals. In particular, the methods proposed in [15,30] consider various real-life cases, but both fall short in real-time processing, a critical aspect for fall detection. Furthermore, due to the usage of neural networks, which require a GPU to accelerate the computation, the approaches from [15,30,33] are relatively difficult to deploy compared with ours.

In contrast, our system achieves high accuracy and precision in both human tracking and fall detection. Additionally, we support scenarios of multiple individuals in both features and provide a fall detection alert service via Gmail. By abstaining from using neural networks and GPU, we enhanced our real-time system to 20 FPS without accuracy drop and ensure ease of deployment for edge platforms. Furthermore, our system offers the flexibility to integrate additional radars for covering blind spots in complex indoor environments if required.

## 7. Conclusions

In this study, we developed a real-time tracking and fall detection system designed for multiple human targets indoors by deploying three mmWave radars developed by TI in a multi-threaded environment. Our discussion delved into how our experimental field setup maximizes the radars’ ability to recognize humans. Additionally, we introduced novel strategies, including dynamic DBSCAN clustering, a probability matrix for tracking updates, target status prediction, and a feedback loop for noise reduction for both the tracking system and fall detection. Our comprehensive evaluation showcases impressive results, achieving 98.9% precision for a single target, as well as 96.5% and 94.0% for two and three targets in human tracking, respectively. For human fall detection, the overall accuracy reached 96.3% with a sensitivity of 99.0% for the fall category, demonstrating the system’s capability to distinguish falls from other statuses. Moreover, we assessed the practicality of fall posture estimation utilizing 3D point clouds. This estimation holds the potential to offer remote medical intervention before on-site medical assistance arrives.

This research lays the groundwork for the development of advanced techniques in human tracking and fall detection using mmWave radar technology. The outcome is a non-intrusive, contactless system featuring real-time processing at 20 FPS on a general-purpose CPU, suitable for applications in industrial and home Internet of Things (IoT) settings. Furthermore, the utilization of lightweight techniques makes it feasible to transfer the system onto low-power-consumption platforms. The success achieved in human detection and tracking opens avenues for future research in more complex HAR tasks using mmWave radars. Future work may involve accommodating larger room sizes, distinguishing between sleeping on the ground and falling, and identifying more postures.

## Figures and Tables

**Figure 1 sensors-24-03660-f001:**
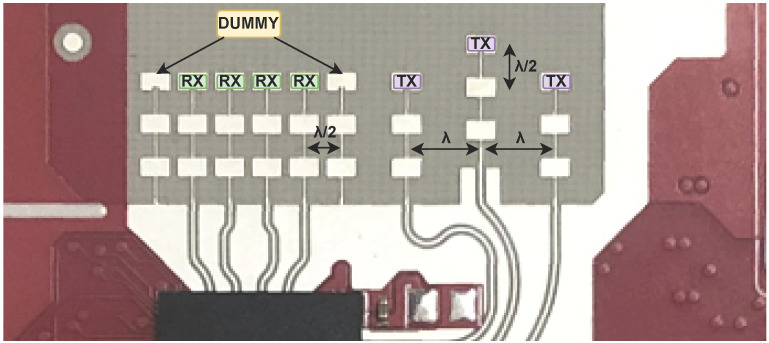
The antenna layout of TI IWR1843 mmWave radar [21].

**Figure 2 sensors-24-03660-f002:**
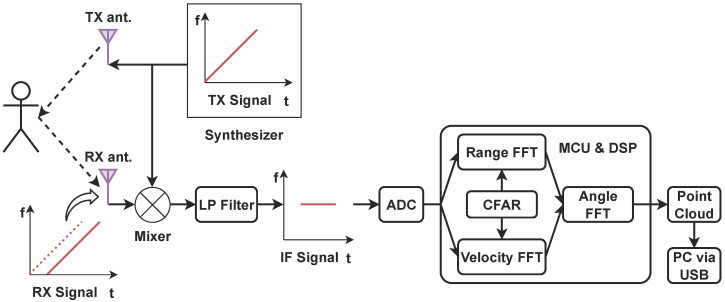
Data processing chain of the TI mmWave radar.

**Figure 3 sensors-24-03660-f003:**
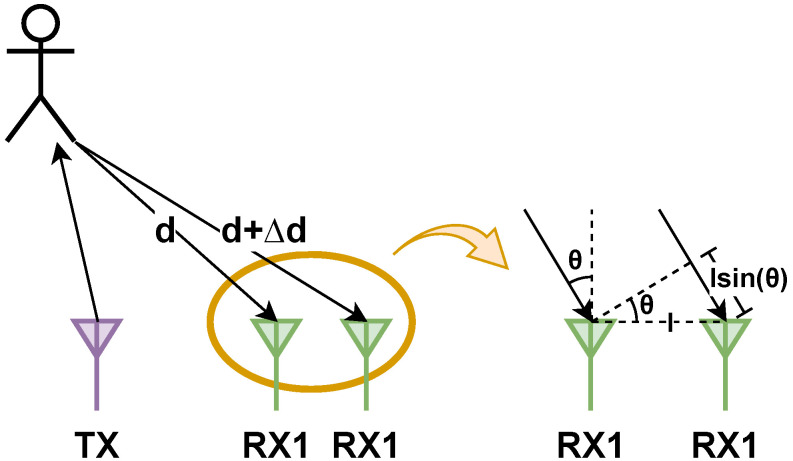
AoA estimation through the utilization of multiple antennas.

**Figure 4 sensors-24-03660-f004:**
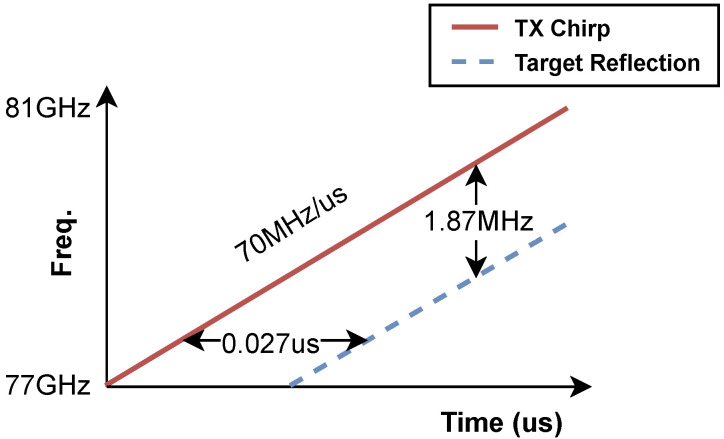
The sending chirp and the reflection chirp from the target.

**Figure 5 sensors-24-03660-f005:**
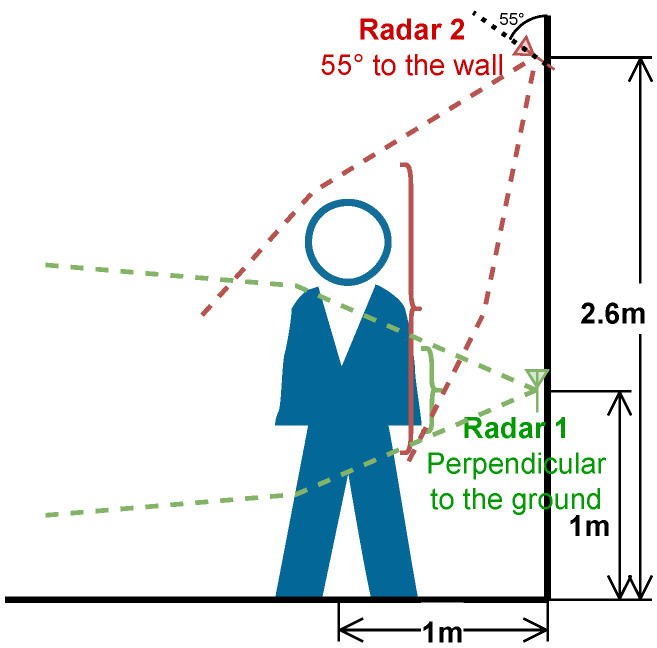
A comparison of the cover area between a radar installed at a height of 1 m (green) and a radar positioned at 2.6 m with 55° to the wall (red).

**Figure 6 sensors-24-03660-f006:**
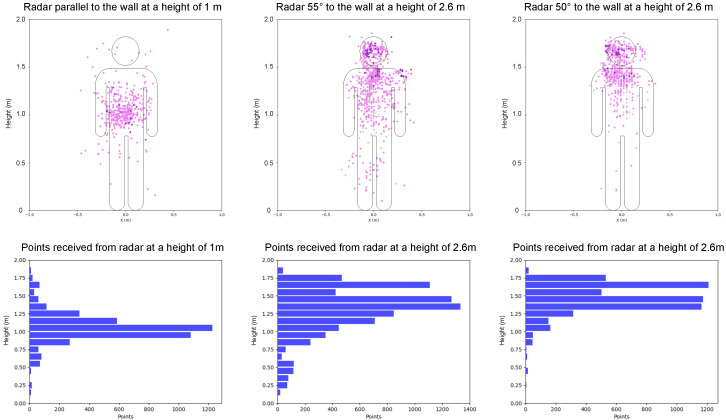
The point clouds acquired from the 1 m height radar (**left**), the 2.6 m height radar with a tilt of 55° (**middle**), and the 2.6 m height radar with a tilt of 50° (**right**), along with the corresponding histograms depicting the number of points recorded over a duration of 10 s when a target was positioned approximately 1 m in front of the radars.

**Figure 7 sensors-24-03660-f007:**
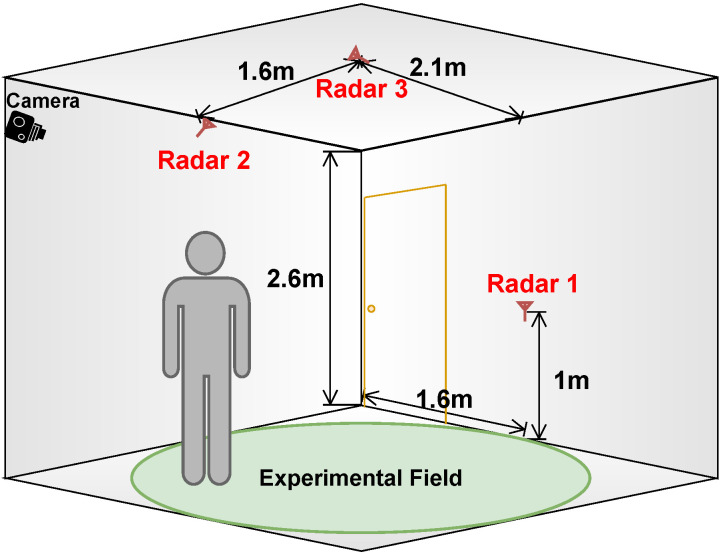
A 3D view simulation with measurement details.

**Figure 8 sensors-24-03660-f008:**
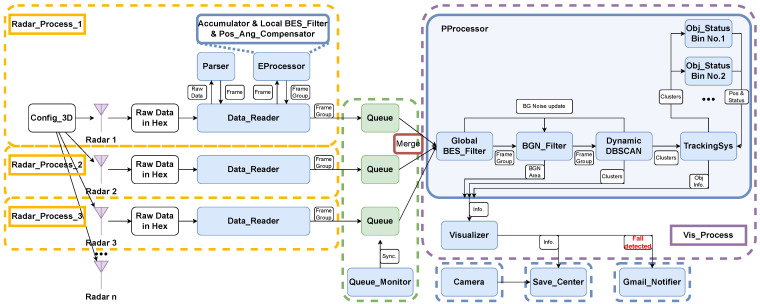
The working flow chart of our software system for real-time human tracking and fall detection. The blue boxes represent each module, while the arrows represent data transfer between modules. Each dotted box donates a concurrent working process.

**Figure 9 sensors-24-03660-f009:**
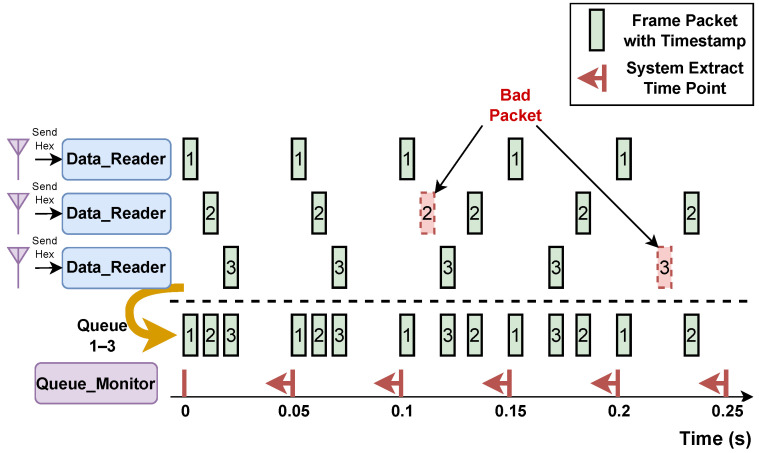
The working flow demonstrates the timeline for the synchronization challenge. Bad packets, highlighted in red, are incomplete packets or corrupted packets, which are discarded. The system is expected to work at 20 frames per second.

**Figure 10 sensors-24-03660-f010:**
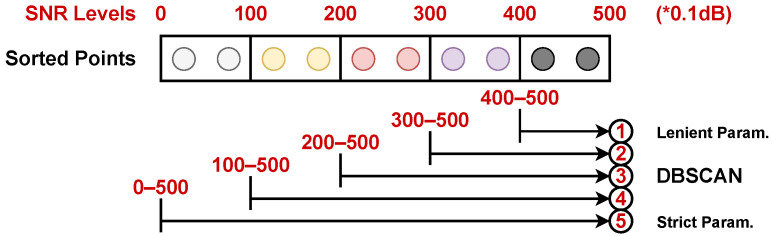
Our dynamic DBSCAN strategy based on the signal SNR level (unit 0.1 dB) performs multiple DBSCAN clusterings for each data frame.

**Figure 11 sensors-24-03660-f011:**
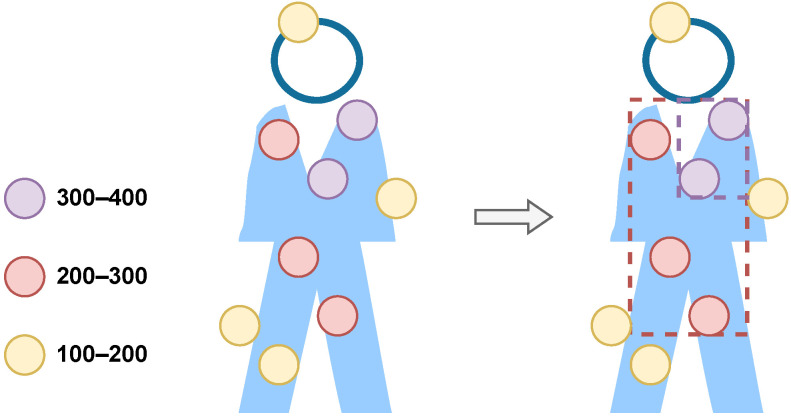
The DBSCAN clusters formed for the point cloud. On the left side, the DBSCAN algorithm was implemented using default parameters, whereas on the right, our strategy of dynamic DBSCAN was applied.

**Figure 12 sensors-24-03660-f012:**
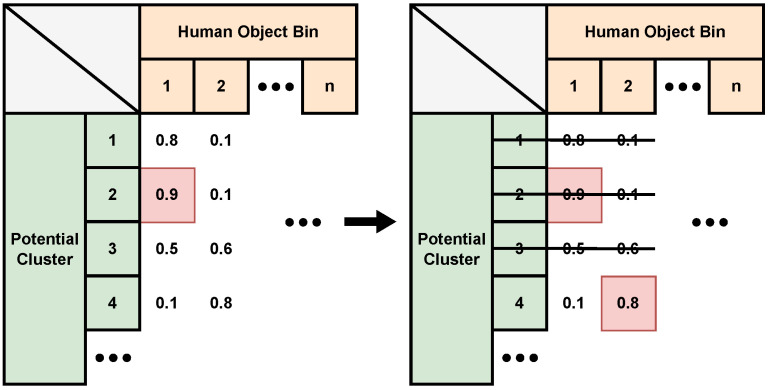
The probability matrix for the *TrackingSys* module. According to the global maximum value selected from the probability matrix, cluster 2 is updated to person 1. Then, cluster 2 and its neighbours 1 and 3 are removed, while cluster 4 is allocated to person 2, as shown on the right.

**Figure 13 sensors-24-03660-f013:**
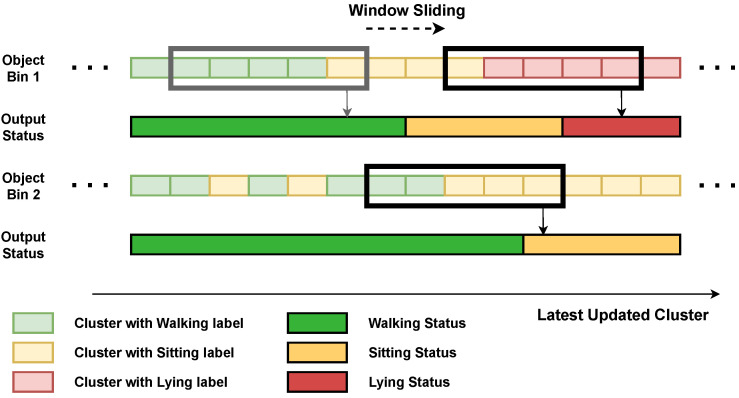
The target status blur process with a sliding window of a certain length (five clusters in this figure).

**Figure 14 sensors-24-03660-f014:**
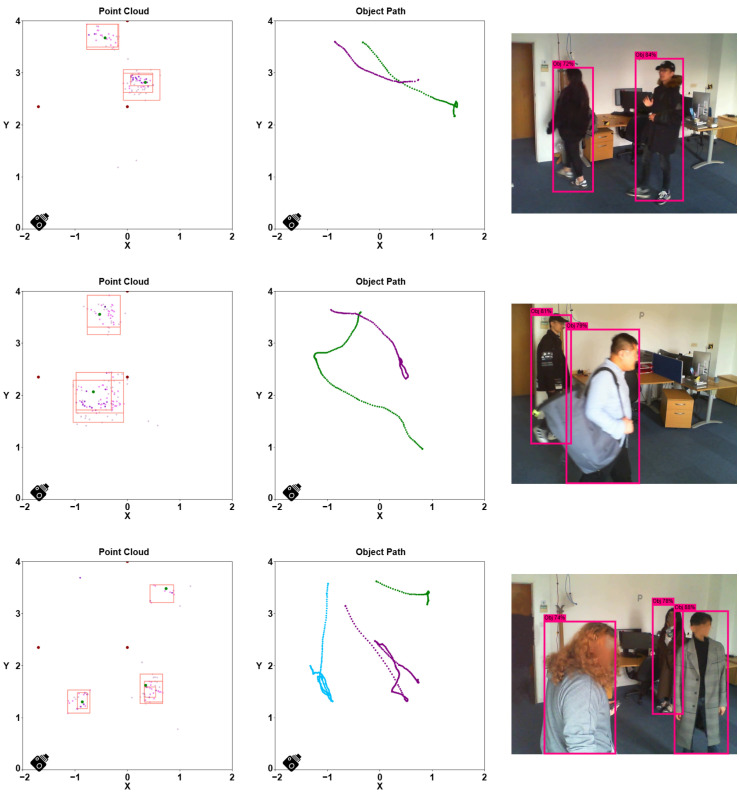
Three scenarios involving multiple individuals in the field are presented, each accompanied by the top view of the point cloud (**left**), the target trajectory (**middle**), and the real image from the camera (**right**). The unit is meters.

**Figure 15 sensors-24-03660-f015:**
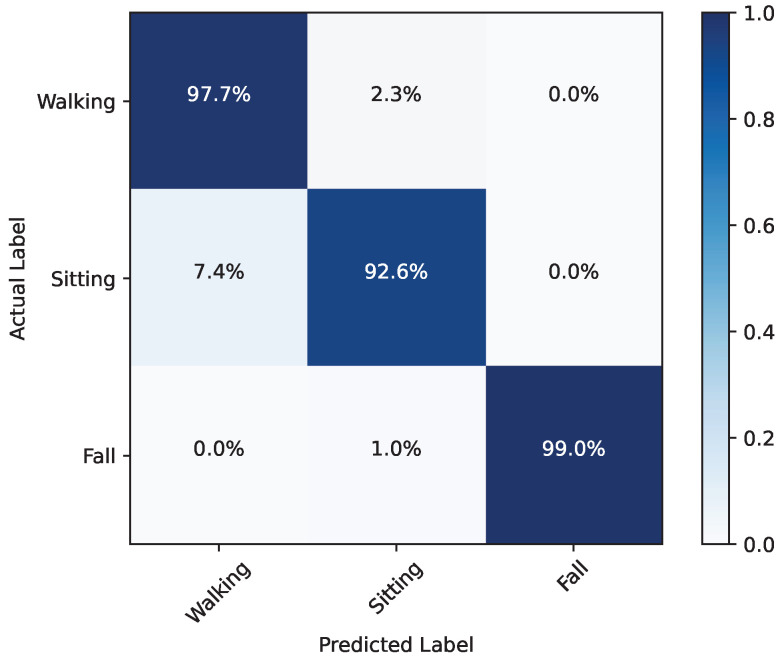
Confusion matrix of three statuses: walking, sitting, and fall detected (lying or sitting on the ground).

**Figure 16 sensors-24-03660-f016:**
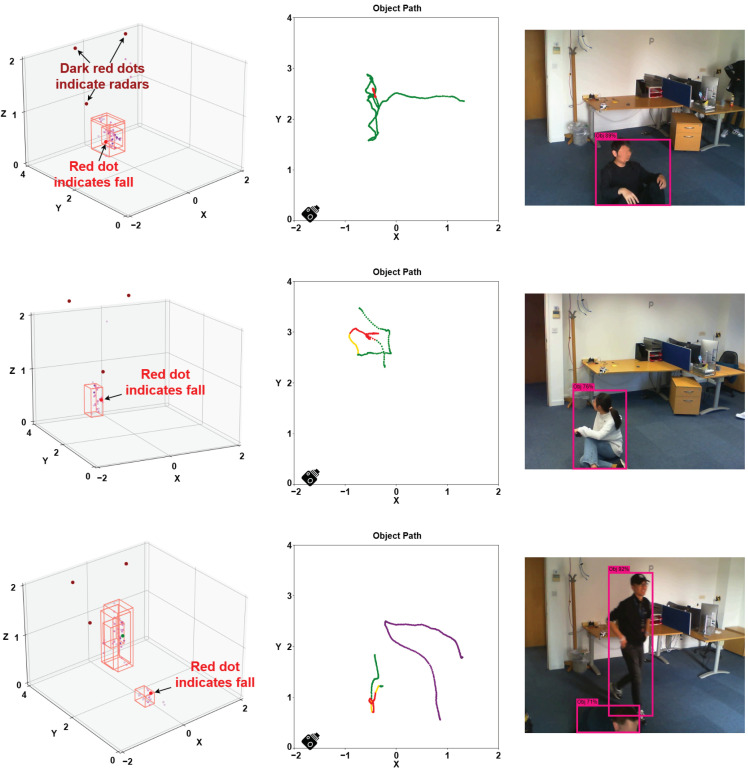
Three scenarios of human fall detection are presented, each accompanied by the 3D view of the point cloud (**left**), the target trajectory (**middle**), and the real image from the camera (**right**). The unit is meters.

**Figure 17 sensors-24-03660-f017:**
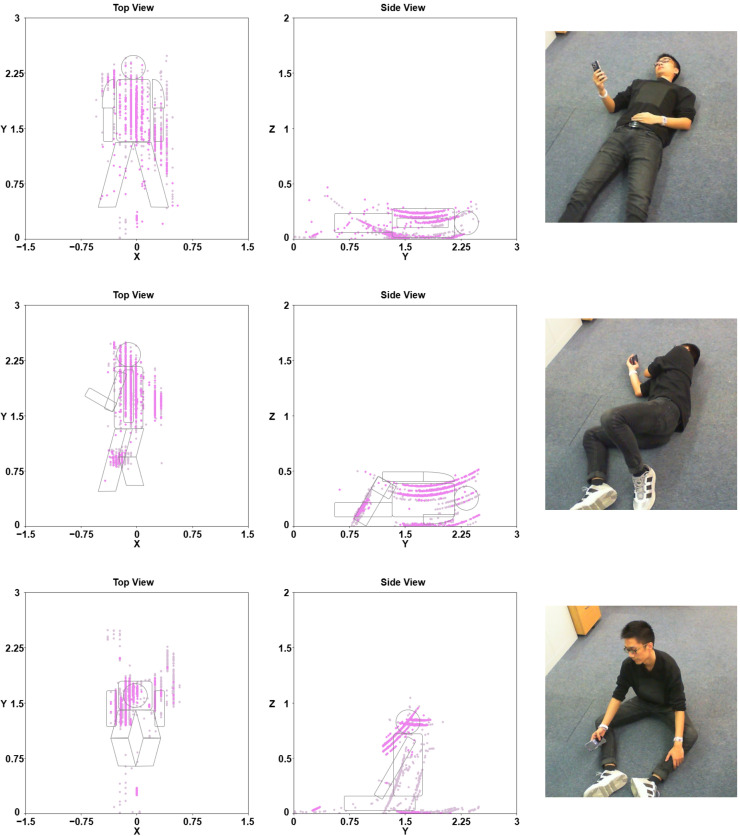
Three fall posture scenarios are presented, each accompanied by point clouds captured from both the top view (**left**) and side view (**middle**) of the point cloud, along with the corresponding ground truth obtained from the camera (**right**). The unit is meters.

**Table 1 sensors-24-03660-t001:** Data collection composition in minutes.

Scenario	Duration	Total Duration	Walking	Sitting	Fall
1 target	156	156 × 1	75.4	32.1	48.5
2 targets	100.4	100.4 × 2	103.3	58.3	39.2
3 targets	46.7	46.7 × 3	77.3	35.7	27.1
Total	303.1	496.9	256	126.1	114.8

**Table 2 sensors-24-03660-t002:** Human tracking performance of our real-time system.

	Sensitivity	Precision	F1 Score
For one target	97.8%	98.9%	98.4%
For two targets	98.2%	96.5%	97.3%
For three targets	97.9%	94.0%	95.9%

**Table 3 sensors-24-03660-t003:** System comparison.

	Wearables [12]	Camera [4]	MmWave Radar [33]	MmWave Radar [15]	MmWave Radar [30]	MmWave Radar (Ours)
Fall Detection	Acc. 93.0%	**Acc. 96.9%**	**Acc. 97.6%**	**Prec. 97.5%**	Acc. 92.3%	**Acc. 96.3%**
Human Tracking	No	Yes	No	No	No	**Prec. 98.9%**
Multiple People	No	No	No	No	No	**Yes**
Fall Detection Alert	**Yes**	**Yes**	No	No	No	**Yes**
Real-time Proc./Speed	Yes	Yes, 8 FPS	Yes, <10 FPS	No	No	**Yes, 20 FPS**
Privacy Concerns	**Low**	Severe	**Low**	**Low**	**Low**	**Low**
Deployment	Inconvenient	Easy, No need GPU	Moderate, Need GPU for NN	Moderate, Need GPU for NN	Moderate, Need GPU for NN	**Easy and Extendable,** **No need GPU**

## Data Availability

Our work can be found at https://github.com/DarkSZChao/MMWave_Radar_Human_Tracking_and_Fall_detection (accessed on 10 March 2024). The dataset is unavailable due to privacy restrictions of ethics.

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
