# Peer review of "Advanced Millimeter-Wave Radar System for Real-Time Multiple-Human Tracking and Fall Detection"

_sensors, 2024, doi:10.3390/s24113660_

Round 1

Reviewer 1 Report

Comments and Suggestions for Authors

In this work the authors present an interesting approach for multiple human tracking and fall-like events detection by mean of 3 FMCW radars. The paper is well organized and clearly written, in my point of view it deserves to be published, however there are some details that may be added and clarified. I hope this will improve the manuscript.

Comments.

·         The novelty of the research should be more clearly stated in the introduction. It should be explicitly indicated which currently unresolved challenges can be addressed with the approaches proposed in the work.

·         P.6 line 176. How and why tilt angle of 550 has been chosen? Were any experiments with other angles conducted for choosing it? With what accuracy must the positioning of a tilted radar be maintained according to the tilt angle so that the effectiveness of the proposed method is maintained?

·         P.7 lines 217-218 It is unclear the location of   ‘… the unmonitored region on the left.’  It is worth to label the areas of potential enter in Fig.7.

·         P.8 Data collection section. A more detailed description of the experiments is necessary in order to assess the conditions under which the proposed method was tested. Has the possibility of simultaneous tracking of several people been tested with a significant difference in their effective scattering surface (short and slender subject and tall or over-weighted)? Were there any experiments where subjects entered simultaneously and moved to some point side by side with a further division of the movement trajectory? Do experiments include stop episodes of one or more subjects for a period of time (for example, as if a person walked into a room and sat down on a sofa)?

·         P.11 How  ‘more lenient DBSCAN parameters’ were chosen? Was it done imperially? Will this set of parameters  be applicable if the proposed approach will be tested in other surroundings (different size of the room and ceiling height)?

·         Fig.12. If two subjects are located in neighboring clusters, will such an algorithm exclude one of subjects and how to avoid this error?

·         P.15 Authors stated that ‘…, we have improved the system’s distinguishability…’, but it is which performance metrics were reached for these difficult cases.

·         P. 15 section 6.2.  The proposed approach makes it possible to detect whether the subject is sitting or lying on the floor, but the authors call this ‘fall detection’, which is not entirely correct. In this case, for example, sleeping on a mattress or sitting at a low table will be mistakenly marked like a fall. In order to assert that the method allows one to detect falls, it is necessary to test it on experimental data with real falls, as was done in similar works.

Author Response

Dear reviewer,

The response is in pdf, see attachment.

Chao

Reviewer 2 Report

Comments and Suggestions for Authors

- Equation 13 suddenly appears and should be preceded by the text

 -  authors could try to better describe how the categorization of the points based on their energy strength is done (what the values mean, is there any normalization etc.)

- a reference to DBSCAN method should be included

Which one of the statements on the AoV the authors state in two different places of the paper is true ? :

a)      according to the radar perspective analysis report in [5], the effective AoV of a single radar is reduced to around ±50 horizontally and ±30 vertically

 b)      However, according to the radar perspective analysis in [5], the effective AoV of a single radar decreases to approximately ±45 horizontally and ±20 vertically due to antenna characteristics and signal attenuation

Author Response

(The authors gave the same response as above.)

Reviewer 3 Report

Comments and Suggestions for Authors

1. Although abbreviations are listed at the end of the paper, it is recommended and always the best practice to elaborate them in their first appearances in the paper followed by using the abbreviations in the rest of the paper.

2. Two contrasting write-up in sec. 3 and sec. 4. Please include just one of them that is accurate.

Sec. 3: line 138-141 The theoretical Angle of View (AoV) for mmWave radars is ±90â—¦ in both horizontal and vertical directions. However, according to the radar perspective analysis report in [5], the effective AoV of a single radar is reduced to around ±50â—¦ horizontally and ±30â—¦vertically due to antenna characteristics and signal attenuation. This reduced vertical AoV means that a radar placed at a height of 1 meter can only capture signals reflected from the human chest to the knee, limiting its ability for complete human body detection.

Sec.4: line 193-196 The mmWave radars theoretically offer an Angle of View (AoV) of ±90â—¦in both horizontal and vertical directions. However, according to the radar perspective analysis in [5], the effective AoV of a single radar decreases to approximately ±45â—¦ horizontally and ±20â—¦ vertically due to antenna characteristics and signal attenuation.

3. Line 202-203, Please discuss the basis of choosing the state and end frequencies, Chirp Cycle Time (Tc), and Frequency Slope, theoretically or experimentally or others?

4. Line 373-379, when the authors mention "cluster", does it mean that even the two- and three-humans are treated as a single object? Whereas from Figure 14, it is not. In such as case, how are the Obj_Status Bin TrackingSys module generate the probability matrix? Pictorial representation similar to Figure 12 is required for two- and three-humans tracking.

5. Line 385-387, how are the human categorized as elderly individuals? Explanation required. What does the authors mean by "Excluding other gestures"? Please include the elaboration.

6. Line 409, what is the unit of measurement of sliding window length?

7. Line 416, how is a frame categorized as a "valid frame"? Please include the explanation.

8. From Table 1, it can be noted that the sensitivity for one- and three-targets is nearly the same, whereas there's a huge difference in the precision. How do you interpret the same? Please include the same in the draft. From the table 1 and the explanation following it, precision is proportion to sensitivity.

9. Figure 14, it will interesting to see the point cloud of the perpendicular camera as well.

10. From the figures of ground truth, I understand the room size is quite small. Please include some experiments in a larger area with objects afar from the radars.

11. Please also cite the below references based on Deep Learning methods alongside [15, 28, 29]

Lin, J.-J.; Guo, J.-I.; Shivanna, V.M.; Chang, S.-Y. Deep Learning Derived Object Detection and Tracking Technology Based on Sensor Fusion of Millimeter-Wave Radar/Video and Its Application on Embedded Systems. Sensors 2023, 23, 2746. https://doi.org/10.3390/s23052746

Wang, T.; Zheng, N.; Xin, J.; Ma, Z. Integrating Millimeter Wave Radar with a Monocular Vision Sensor for On-Road Obstacle Detection Applications. Sensors 2011, 11, 8992–9008.

12. Please include the specifications of the system used for implementation.

Comments on the Quality of English Language

Line 515: Correction needed: "In contrast, our system achieves in achieving high accuracy and precision in both ..."

Author Response

Dear reviewer,

The response is in pdf, pls see attachment.

Chao
